

# A comparative study of different variable selection methods based on numerical simulation and empirical analysis

Dake Hou[1], Wenli Zhou[2], Qiuxia Zhang[3], Kun Zhang[3] and Jiaqi Fang[2]

[1] School of Mathematics, Shandong University, Jinan, China
[2] School of Business, Wenzhou Univeristy, Wenzhou, China
[3] School of Economics and Management, Beijing Jiaotong University, Beijing, China

## ABSTRACT

This study employs the principles of computer science and statistics to evaluate the efficacy of the linear random effect model, utilizing Lasso variable selection techniques (including Lasso, Elastic-Net, Adaptive-Lasso, and SCAD) through numerical simulation and empirical research. The analysis focuses on the model's consistency in variable selection, prediction accuracy, stability, and efficiency. This study employs a novel approach to assess the consistency of variable selection across models. Specifically, the angle between the actual coefficient vector $\beta$ and the estimated coefficient vector $\hat{\beta}$ is computed to determine the degree of consistency. Additionally, the boxplot tool of statistical analysis is utilized to visually represent the distribution of model prediction accuracy data and variable selection consistency. The comparative stability of each model is assessed based on the frequency of outliers. This study conducts comparative experiments of numerical simulation to evaluate a proposed model evaluation method against commonly used analysis methods. The results demonstrate the effectiveness and correctness of the proposed method, highlighting its ability to conveniently analyze the stability and efficiency of each fitting model.

## INTRODUCTION

Statistical modeling aims to attain two objectives: namely, the attainment of a high level of prediction accuracy and the establishment of robust inferential and explanatory capabilities of the model. The classical linear statistical model is a highly effective and extensively utilized statistical technique that retains superior inferential capabilities over nonlinear models in addressing real-world problems. Several novel approaches, including polynomial regression, spline regression, local regression, generalized additive model, and partial linear model, have been developed based on this model, which can be viewed as its advancement and expansion. The process of selecting variables is a crucial aspect of modeling, and the comparative evaluation of its diverse techniques is a current area of focus within the realm of statistics research.

Corresponding author
Jiaqi Fang, 20210634@wzu.edu.cn

*Tibshirani (1996)* introduced the Least Absolute Shrinkage and Selection Operator (Lasso) penalty-based variable selection method, which has been widely adopted due to its ability to enhance model prediction and inference by simplifying the model. Longitudinal data with random effects are a crucial type of data in various fields such as biomedicine, clinical trials, meteorological observation, industrial engineering, and e-commerce platforms. Consequently, it is imperative to undertake comprehensive research on Lasso techniques for variable selection within the context of linear random effect models. Additionally, novel approaches for conducting comparative analyses of multiple variable selection methods should be investigated. Over the last decade, numerous studies have been conducted both domestically and internationally on the theory and practical implementation of the Lasso regression method. The Adaptive Group Lasso method was investigated by *Wang & Leng (2008)*, while the Graphical Lasso method was explored by *Yuan & Lin (2007)*. *Wang & Yin (2008)* introduced the sMave method and its corresponding iterative algorithm in their investigation of the single index model. *Zeng, He & Zhu (2012)* explored the estimation and variable selection of the single index model, utilizing the penalty partial derivative and proposing the sim-LASSO method. This method effectively eliminates the impact of partial derivative points equaling zero on estimation while simultaneously compressing coefficients. In the field of high-dimensional and ultra-high-dimensional models, a comprehensive review of numerous research findings was conducted by *Niu & Li (2021)*. In their research, *Fan & Lv (2008)* examined the utilization of an ultra-high dimensional variable selection method in the generalized linear model. They introduced the sure independence screening (SIS) method, which employs a marginal screening approach to assess the significance of each covariate based on the Pearson correlation between the response variable and a single covariate. This method enables preliminary dimensionality reduction, reducing the number of covariates to a level where traditional penalty variable selection methods can be effectively applied. *Yuan & Lin (2007)* conducted a systematic investigation into the correlation between Lasso and Bayes (*Fan & Lv, 2008*). They employed the LARS algorithm to accomplish the computation of Bayes posterior distribution in scenarios involving a high number of dimensions. The approach of maximizing marginal likelihood was employed and presented a viable technique for the selection of Lasso's penalty parameter (*Yuan & Lin, 2005*). In her study, *Jing (2013)* employs the inverse Bayes formulae (IBF) to develop two novel algorithms that rely on non-iterative sampling techniques. These algorithms demonstrate a rapid and efficient solution to the Bayesian Lasso problem (*Jing, 2013*). In 2019, *Li (2020)* made enhancements to the Bayesian Lasso estimation technique and introduced the Bayesian Adaptive Lasso estimation method, which is tantamount to adaptive Lasso. To assess the inferential explanatory capability of the model, numerous studies in this domain measure the coherence of model variable selection through the computation of true positive rate (sensitivity) and true negative rate (specificity) (*Trevor, Robert & Martin, 2020*; *Wood, 2018*; *Wang & Li, 2018*; *Wu, 2015*; *Gao, 2015*), or the likelihood of predictive variable inclusion in the model (*Liang & Feng, 2016*; *Rodgers & Nicewander, 1988*; *Xu, 2022*). The evaluation of the Lasso variable selection method's model quality was conducted in literature through the utilization of cc, which represents the average number of correctly

selected predictive variables, cd, which denotes the average number of correctly eliminated unrelated predictive variables, and the accuracy of coefficient estimate $\hat{\beta}$ (*Siwei, 2021*; *Lei, 2022*). The quality of the fitted model was assessed in Literature through the comparison of PMSE, which stands for "the test mean square error of the model", IC, which represents "the number of non-zero coefficients incorrectly identified as zero", and C, which denotes "the number of zero coefficients correctly identified as zero" (*Xu, 2022*; *Yanbiao, 2011*). The application of SIS involves the reduction of dimensionality of high-dimensional data with varying correlation coefficients, followed by the utilization of distinct Lasso variable selection techniques for fitting purposes (*Yuan & Lin, 2005*; *Luo, 2021*; *Gao, 2011*). The assessment of the fitted model's quality involves a comparison of the precision of the regression coefficient $\hat{\beta}$ for each model.

This study presents a novel approach to assess the consistency of variable selection in regression models by calculating the Angle between the "true coefficient vector" $\beta$ and the "estimated coefficient vector" $\hat{\beta}$. The Angle is then used to evaluate the accuracy of model predictions and is visualized using boxplots. The distribution of Angle and Err (mean square error) is presented to provide an intuitive understanding of the data. Additionally, the stability of each model is compared based on the number of abnormal values. The present study employs a methodology that integrates numerical simulation and empirical analysis to compare and scrutinize the Lasso variable selection techniques. Concurrently, numerical simulation experiments are conducted to juxtapose the model evaluation technique posited in this manuscript with the conventional analytical methodologies. The findings of the experiment indicate that the outcomes of the model derived in this investigation are in agreement with the established theory and the pertinent research in this domain. This approach exhibits not only accuracy and efficacy, but also the advantageous attributes of facilitating the evaluation of stability and effectiveness of individual fitting models. The present study utilizes median values for the model performance indicators Angle and Err, in contrast to the commonly employed index average value found in the aforementioned literature. This approach effectively mitigates the influence of outliers in the data on statistical analysis, thereby enhancing the precision of the fitted model's performance evaluation. The present study employs the ten-fold cross validation technique to obtain the penalty parameter $\lambda$ for the model. As per the suggestion put forth by *Fan & Li (2001)*, the value of parameter $\alpha$ in SCAD is 3, whereas the weight coefficient $\gamma$ in Adaptive Lasso is 1. Further investigation is required to establish the parameters in the penalty function, taking into account the diverse data characteristics and application background conditions of the model. Secondly, with regards to the prevalent issue of data contamination in the current gathered datasets, it is imperative to conduct further research on the resilience of the model.

## MODEL AND VARIABLE SELECTION METHODS

A classical linear regression model is described as:

$$y_i = \beta_0 + \sum_{j=1}^{p} x_{ij}\beta_j + e_i \qquad i = 1, 2, \ldots, n \tag{1}$$

There are $n$ groups of observations, each of which consists of an output variable $y_i$ and $p$ associated predictive variables $x_i = (x_{i1}, x_{i2}, \ldots, x_{ip})^T$. $\beta_0$ and $\beta = (\beta_1, \beta_2, \ldots, \beta_p)^T$ are unknown parameters and $e_i$ is the error term. The objective of linear regression is to forecast the response variable based on the explanatory variables and to determine the significance of the explanatory variables. The primary task involves the estimation of the unknown parameter $\beta$. The conventional approach involves minimizing the objective function's least square method.

$$\underset{\beta_0, \beta}{minimize} \sum_{i=1}^{n} \left( y_i - \beta_0 - \sum_{j=1}^{p} x_{ij} \beta_j \right)^2 \qquad (2)$$

In Eq. (2), it is generally observed that none of the least squares estimates of $\beta$ are equivalent to zero. When the value of $p$ is high, the ultimate model becomes challenging to elucidate. Additionally, in cases where $n$ is less than $p$, the outcomes of the least squares estimations are non-unique, and there exist an infinite number of solutions that can render the objective function to be equivalent to zero. Hence, it is imperative to restrict (regularize) the process of estimation by incorporating a penalty function. The penalty function can be categorized into two types based on the degree of concavity, namely convex penalty (*e.g.*, Lasso, Adaptive Lasso) and non-convex penalty (*e.g.*, SCAD, MCP, *etc.*).

The convex function is defined on a convex set $S$ which is meet the conditions: $\forall x_1, x_2 \in S, \forall a \in [0, 1]$, there is

$\quad f(ax_1 + (1-a)x_2) \leq af(x_1) + (1 - a)f(x_2)$.

(*Definition of $S$*: for $\forall x_1, x_2 \in S, \forall a \in [0, 1]$, *when* $x = ax_1 + (1-a)x_2, x \in S$)

The utilization of convex penalty can guarantee the singular nature of the solution. Additionally, proficient algorithms exist to obtain estimators that exhibit favorable attributes of stability and sparsity. However, it is important to note that these estimators are only partial in nature and lack Oracle properties.

The utilization of the non-convex penalty can effectively achieve coefficient sparsity while satisfying the Oracle properties. However, due to its non-convex nature, the uniqueness of solutions cannot be guaranteed, resulting in the emergence of multiple local optima. Consequently, the stability of the results may be relatively poor. Furthermore, the inclusion of a concave parameter in the penalty function results in heightened computational complexity.

The penalty function can be categorized into two types, namely the Penalized Residual Sums of Squares and the penalty likelihood function, based on the specific models employed. In the context of a general linear regression model (LM) with a normally distributed dependent variable, an identity joining function, and a loss function that involves the sum of squares of residuals, the penalty residuals are utilized to calculate the sum of squares. The penalty likelihood function can be utilized in various statistical distributions, including the Poisson distribution, binomial distribution, and gamma distribution of dependent variables. Additionally, it can be applied to the exponential cluster of connection function. The loss function is a generalized linear model of the likelihood function, which encompasses models such as the logistic regression model and the Poisson regression model.

With parameter set $\beta = (\beta_1, \beta_2, \ldots, \beta_p)^T$ as the study object, on the basis of *OLS*, increase the constraint conditions about the $\beta$, the penalty function $p_\lambda(|\beta|)$, thus, to establish the punishment least squares estimation (*Penalized Least Squares, PLS*)

$$\hat{\beta} = \operatorname{argmin}\left\{ \sum_{i=1}^{n}\left(y_i - \beta_0 - \sum_{j=1}^{p} x_{ij}\beta_j\right)^2 + \sum_{j=1}^{p} p_\lambda(|\beta_j|)\right\} \tag{3}$$

Different variable selection methods can be constructed when we select different penalty functions.

## Lasso method and basic characteristics

*Tibshirani (1996)* introduced a technique for selecting variables. The fundamental concept underlying Lasso involves the imposition of penalty factors to restrict the $L_1$ norm of the estimator $\beta$, building upon the framework of Ordinary Least Square (OLS) estimation (*Tibshirani, 1996*).

As for the data set with $p$ predictive variables and $n$ predictive variable-response variable pairs $\{(x_i, y_i)\}_{i=1}^{n}$, we could use Lasso to find an estimate of $\hat{\beta}$ that can better fit the data through minimization of $RSS(\hat{\beta})$.

$$RSS(\hat{\beta}) = \underbrace{\operatorname{argmin}}_{\hat{\beta}} \sum_{i=1}^{n}\left(y_i - \sum_{j=1}^{p}\beta_j x_{ij}\right)^2 + \lambda\sum_{j=1}^{p}|\beta_j| \tag{4}$$

$\lambda \geq 0$ is the Penalized Parameter, $\lambda\sum_{j=1}^{p}|\beta_j|$ is compression constraints.

$$\hat{\beta} = \operatorname{argmin}\left\{\|y - x\beta\|_2^2\right\} s.t\, \|\beta\| \leq t. \tag{5}$$

$t \geq 0$ is a tuning Parameter, which controls the intensity of compression. If the Parameter obtained by the least squares is estimated as $\beta^0$, Lasso can be compression as long as $t < \sum_{j=1}^{p}\left|\widehat{\beta_j^0}\right|$. In addition, for some models with small absolute values, the coefficient is compressed to zero. Therefore, the inequality $\|\beta\| \leq t$ effectively limits the parameter space and makes the final model explicable.

It can be proved that the relation between $t$ and $\lambda$ (*Wu, 2015*) is

$$t = \sum_{j=1}^{p} sign(\widehat{\beta_j^0}) \cdot \widehat{\beta_j^0} - p\frac{\lambda}{2}. \tag{6}$$

The estimator $\hat{\beta}$ can be computed through numerical approximation techniques, such as coordinate descent and minimum angle regression.

The fundamental attributes of Lasso entail pursuing the sparse representation of the model, essentially. The aforementioned procedure is executed through the optimization of a function problem that comprises a combination of a loss and a penalty term. In cases where a set of predictive variables exhibit high correlation, Lasso regression may select one variable while disregarding the others, leading to unstable outcomes. This indicates that collinearity cannot be effectively and accurately addressed.

## Adaptive Lasso method and basic characteristics

It can be seen from Lasso analysis that Lasso penalty function is the same for all the estimators in $\beta = (\beta_1, \beta_2, \ldots \cdots, \beta_p)$. However, if the penalty function can be made to give a smaller penalty to the larger estimators, thus effectively reducing the model's bias while ensuring the sparsity of the model. Therefore, *Zou (2006)* proposed Adaptive Lasso, in which the punishment parameter can be adjusted according to the size of the estimator, instead of $\lambda$ being a fixed punishment in Lasso (*Liang & Feng, 2016*). Before the Adaptive Lasso in $\|\beta_j\|$ increase weight coefficient $\hat{w}_j = \frac{1}{|\hat{\beta}_j|^\gamma}, \gamma$ constant is greater than zero, Zou advised to choose a $\sqrt{n}$ consistent estimator satisfy, which uses least squares estimator $\hat{\beta}$ as the initial estimator, $\hat{\beta}_j = \hat{\beta}^{\mathrm{OLS}}$, Adaptive Lasso is in the form of

$$\hat{\beta} = \underset{\beta}{\arg\min} \|y - X\beta\|_2^2 + \lambda \sum_{j=1}^{p} \hat{w}_j \|\beta_j\|_1. \tag{7}$$

By formula Eq. (7), when $\gamma \geq 1$, if some components of the initial estimate of $\hat{\beta}$ are large (such as $\hat{\beta}_j$), the weight of the corresponding punishment item weight coefficient $\hat{w}_j = \frac{1}{|\hat{\beta}_j|^\gamma}$ relative to other components, so the Adaptive - Lasso has the following features:

(1) The utilization of a penalty function serves to decrease the bias of the Lasso estimator, while simultaneously guaranteeing the accuracy of the estimator.

(2) It also proves that Adaptive Lasso can satisfy the following Oracle properties with an appropriate $\lambda$ value:

(a) The consistency of variable selection:
$$\{j, \hat{\beta}_j \neq 0\} \;=\; \{j, \beta_j \neq 0\} \;\triangleq\; S_0.$$

(b) Asymptotic normality:
$$\sqrt{n}(\hat{\beta} - \beta_{S_0}) \xrightarrow{d} N(0, \sigma^2).$$

$^*$ $S_0$ is the active set of predictive variables
$$S_0 = \left\{j : \beta_j^0 \neq 0, j = 1, 2, \ldots\ldots, p\right\},$$
$$\hat{S}(\lambda) = \left\{j : \hat{\beta}_j(\lambda) \neq 0, j = 1, 2, \ldots\ldots, p\right\},$$
$\hat{S}(\lambda)$ Is a nonzero coefficient subscript set of parameters estimated by Lasso method.

According to the *CV* method to get $\hat{\lambda}_{CV}$, eventually get $\hat{S}(\hat{\lambda}_{CV})$ which has a high probability to include $S_0$ and $\left|\hat{S}(\hat{\lambda}_{CV}) \leq \min(n, p)\right|$.

(3) The Adaptive Lasso method conforms to the three characteristics of the penalty function introduced by Fan and Li, namely impartiality, parsimony, and continuity. These attributes are an advancement over the Lasso method. However, the Adaptive Lasso method still encounters challenges in effectively addressing collinearity.

## Elastic Net method and basic features

The Ridge regression technique is distinguished by its ability to evenly allocate weights to associated characteristic variables. Conversely, the Lasso method is not capable of effectively addressing collinearity. In response, it introduced the Elastic Net approach, which combines the strengths of both Ridge and Lasso regression techniques. Specifically, Elastic Net is a convex combination of Ridge and Lasso regression (*Rodgers & Nicewander,*
*1988*).

$$\hat{\beta} = arg \min_{\beta \in R^p} \left\{ \|y - X\beta\|_2^2 + \lambda_1 \sum_{j=1}^{p} |\beta_j| + \lambda_2 \sum_{j=1}^{p} \beta_j^2 \right\} \tag{8}$$

The penalty term is $P_\lambda\left(|\beta_j|\right) = \lambda_1 |\beta_j| + \lambda_2 |\beta_j|^2$, $\lambda_1, \lambda_2$ are two nonnegative punishment parameters, order $a = \frac{\lambda_2}{\lambda_1 + \lambda_2}$,

$$\hat{\beta} = arg \min_{\beta \in R^p} \left\{ \|y - X\beta\|_2^2 + (1-a) \sum_{j=1}^{p} |\beta_j| + a \sum_{j=1}^{p} \beta_j^2 \right\} \tag{9}$$

Penalty term $P_\lambda\left(|\beta_j|\right) = (1-a)\sum_{j=1}^{p} |\beta_j| + a\sum_{j=1}^{p}\beta_j^2$ at zero point is the guide, for $\forall a > 0$ is strictly convex.

To: $X^*_{(n+p) \times p} = \frac{1}{\sqrt{1+\lambda_2}} \left( \begin{smallmatrix} X \\ \sqrt{\lambda_2}I \end{smallmatrix} \right)$,    $Y^*_{n+p} = \left( \begin{smallmatrix} Y \\ 0 \end{smallmatrix} \right)$,

$\gamma = \frac{\lambda_1}{\sqrt{1+\lambda_2}}$,     $\beta^* = \sqrt{1+\lambda_2}\beta$,

then

$\hat{\beta}^* = arg\min_{\beta^* \in R^p} \left\{ \|y^* - X^*\beta^*\|_2^2 + \gamma \sum_{j=1}^{p} \left|\beta_j^*\right| \right\}$.

Therefore, the Elastic Net problem can be converted to the Lasso problem. The optimal Elastic Net solution $\hat{\beta}^*$ can be found on $\hat{\beta} = \frac{1}{\sqrt{1+\lambda_2}}\hat{\beta}^*$. Because $I_{p \times p}$ is full rank, considering the composition of $X^*_{(n+p) \times p}$ and it has $p$ columns, the Lasso matrix $X^*_{(n+p) \times p}$ (matrix rank $p$), the applied matrix is $X_{(n+p) \times p}$. Therefore, Elastic Net can select up to $p$ variables which solved Lasso can only select a maximum of $n$ ($n < p$) variables. According to the findings, the Elastic Net technique is capable of selecting group variables, meaning that it can effectively select multiple highly correlated independent variables. This addresses the issue commonly encountered with Lasso regression methods, which tend to select only one variable and disregard the others. This approach is appropriate for certain application scenarios that require analysis of the correlation between dependent variables and predictive variables exhibiting group characteristics. Examples of such scenarios include variable screening and prediction of gene expression profile data.

## SCAD method and basic features

In their study, *Fan & Li (2001)* introduced a non-convex penalty function with the aim of attaining unbiasedness of the estimator $\beta$. This penalty function ensures that the penalty on the coefficient decreases as the coefficient estimator increases, thereby ensuring the approximate unbiasedness of large coefficients.

$$\hat{\beta} = arg \min_{\beta \in R^p} \left\{ \|y - X\beta\|_2^2 + \sum_{j=1}^{p} P_\lambda\left(|\beta_j|\right) \right\}, \tag{10}$$

$P_\lambda(|\beta_j|)P$ is the penalty term. The proposed method is a continuously differentiable penalty function, called smooth cohesion absolute deviation penalty SCAD (*Smoothly*

*Clipped Absolute Deviation Penalty*).

$$P\left(\beta_j|_{\lambda,a}\right) = \begin{cases} \lambda|\beta_j| & |\beta_j| \le \lambda, \\ \dfrac{2a\lambda|\beta_j| - |\beta_j|^2 - \lambda^2}{2(a-1)} & \lambda < |\beta_j| < a\lambda, \\ \dfrac{(a+1)\lambda^2}{2} & |\beta_j| \ge a\lambda, \end{cases} \tag{11}$$

Take the derivative of the penalty function *SCAD*, get

$$P'\left(\beta_j|_{\lambda,a}\right) = \begin{cases} sgn(\beta_j)\lambda & |\beta_j| \le \lambda \\ sgn(\beta_j)(\alpha\lambda - |\beta_j|)/(\alpha-1) & \lambda < |\beta_j| < a\lambda \\ 0 & |\beta_j| \ge a\lambda \end{cases} \tag{12}$$

Thus, its coefficient estimate speed of punished decreases with the coefficient estimators $|\beta_j|$ increases, namely when the $\beta_j$ absolute value is larger, SCAD penalty is constant, when the $\beta_j$ absolute value is small, the smaller the absolute value of the coefficient of the degree of compression than the LASSO method is bigger and therefore better able to induce the sparse structure, $t$ is convenient for screening and obtaining a sparse subset of variables. Therefore, SCAD guarantees the unbiasedness of large coefficients. Not differentiable at the origin, but continuously differentiable at $(-\infty,0) \cup (0,+\infty)$, a local quadratic approximation is used to obtain a local optimal solution while ensuring sparsity and continuity. Fan, Huang, and Kim respectively proved that N-SCAD (conventional), H-SCAD (high dimensional), and UH-SCAD (super high dimensional) have Oracle properties under certain hypothetical conditions, and have the advantage over Adaptive Lasso in that it does not require prior access to the consistent estimator of $\sqrt{n}$ to $\hat{\beta}^0$. It is important to note that the local optimal solution may not always coincide with the global optimal solution. Numerical methods often yield a particular local solution, thereby creating a disparity between theoretical and practical applications. Consequently, research in this area remains a prominent topic of interest.

## Numerical simulation and comparative analysis

This study conducts a comparative analysis of the penalty function of Lasso, Adaptive Lasso, Elastic-Net, and SCAD variable selection methods under varying conditions through numerical simulation. The study also examines the applicable application scenarios of these methods.

The model's performance was assessed using four indicators, namely: (1) the consistency of model variable selection, (2) the mean square error, (3) the efficiency of the model algorithm, and (4) the stability of the model. Regarding the evaluation of the consistency of model variable selection, the false discovery rate and false exclusion rate methods as well as the calculation of the probability of predictive variables entering the model were not utilized in our approach (*Jing, 2013*; *Wood, 2018*; *Xu, 2022*; *Min, 2022*; *Chu, 2018*; *Hastie, Tibshirani & Wainwright, 2015*; *Bai & Zhao, 2005*). However, we choose variable selection compatibility of the model to be found through calculating the Angle between uniformed estimated coefficient vector $\hat{\beta}$ and the uniformed true coefficient vector $\beta$ Angle $= \frac{180}{\pi}\arccos\left(\beta^T\hat{\beta}\right)$ reflection coefficient estimation accuracy, namely consistency

$(\{j : \widehat{\beta_j} \neq 0\} = \{j : \beta_j \neq 0\} \triangleq S_0; S_0$ is the active set of predictive variables). The Angle is closer to zero, the higher the degree of compatibility between $\beta$ and $\hat{\beta}$, the stronger the model interpretation ability (*geometric interpretation of Pearson correlation coefficient formula*) (*Liang, Feng & Song, 2016*). $Err = \frac{1}{n} \sum (\hat{y}_0 - y_0)^2$, the mean square error of the predicted variable obtained from the training set which can be carried to the model, the mean square error of the response variable $\hat{y}_0$ and the actual response value $y_0$ in the test set can reflect the prediction accuracy of the model. If the predicted response value is close to the actual response value, $\Delta Err$ will be small; the stability of the model was assessed by utilizing the boxplot tool to examine the outliers of Angle and Err across various variable selection methods. This approach provided insight into the robustness of the established model. The evaluation of the model algorithm's efficiency is based on the duration of time taken by the computer to perform a regression fitting.

Four indicators were used to evaluate the model performance:
Angle:consistency of model variable selection;
    —- Evaluate the explanatory power of the model
Err:mean square error;
    —- Evaluate the prediction accuracy of the model
Outliers:Evaluate the stability of model
Time:Evaluate the efficiency of model algorithm

## Design of simulated data set

Our data are generated from the following linear random effects model

$$Y = X\beta + Z\delta + \varepsilon \qquad (13)$$

$y_{n \times 1}$ is the response variable, $X_{n \times p}$ is the design matrix of the prediction variable, $Z_{n \times q}$ is the design matrix of the random effect variable, which is constructed in the same way as X, $q \leq p.\beta_{p \times 1}$ is a nonrandom parameter vector called deterministic effect, $\delta_{q \times 1}$ is a random parameter vector called random effect, $\varepsilon_{n \times 1}$ is a random noise vector. $\delta$ follows the normal distribution with zero mean and $G$ covariance matrix. $\delta \sim N(0, G)$, $\varepsilon$ follows normal distribution, $\varepsilon \sim N(0, R)$, $R = \sigma^2 I_n$, $cov(\delta, \varepsilon) = 0$. In order to design the simulation data set, the function mvrnorm() in MASS package is firstly used to design the $n \times p$ dimensional multivariate normal simulation matrix about the prediction variable X, $X_{n \times p}$ is the prediction variable, let $\beta_{p \times 1} = (1, 1.2, 3.2, 4.1, -5, \underbrace{0, \ldots 0}_{45})$- is the coefficient vector, and $y_{n \times 1}$ is the response variable. The model discussed is the sparse model, $\varepsilon_{n \times 1}$ is the random noise vector, which is independently identically distributed and follows the standard normal distribution, $\varepsilon_{n \times 1} \sim N(0, I_n)$, X and $\varepsilon$ are independent. The numerical simulation experiment was repeated for 300 times, and the data in the X matrix was updated each run. In this study, the mean value of the prediction variables in each column of the X matrix was designed to be a random variable $mu \sim N(0, I_n)$ subject to normal distribution instead of constant, to simulate the internal fluctuation of the prediction variables and the non-homogeneity of the random factors $Z\delta$ between individuals. The fitted model has linear random effects. The extent of collinearity exhibited

by the predictor variable $X$ is strongly associated with the correlation coefficient that exists between them. In order to assess collinearity, experimental designs are implemented that incorporate correlation coefficients. A correlation coefficient of 1 corresponds to complete collinearity, while a coefficient of 0 indicates the absence of collinearity. As the correlation coefficient approaches 1, the degree of collinearity between variables increases. The dataset was partitioned into two subsets, namely the training set and the test set. Four variable selection techniques, namely Lasso, Elastic-Net, SCAD, and Adaptive Lasso, were employed to perform regression modeling on the dataset. The evaluation model was based on the average value of four indicators obtained from conducting a numerical simulation experiment for a total of 300 times.

The numerical simulation is carried out in the following two scenarios.

(1) $n \gg p$, where the number of observed samples n is much larger than the conventional data set with the number of predictive variables P, n:p = 2500:50; The correlation coefficient between the predictive variables was cor = 0.2/0.4/0.6/0.8/0.98.

(2) high dimensional data set which satisfying $log(p) = n^a (0 < a < 1)$

n:p =40:50; Correlation coefficient cor =0.2/0.4/0.6/0.8/0.98

We use R to analyze and complete numerical simulation, mainly using the following software packages that MASS, car, psych, glmnet, ncvreg, msgps, ISLR *etc.* We select parameters in software analysis:

For SCAD, we used ncvreg(), where the gamma parameter selection is 3.7 (*i.e., the punishment function where a is recommended by Fan*); alpha $= 1$ (*MCP/SCAD penalty*); When n<p, lambda.min is set to 0.05 instead of 0.001;

For Adaptive Lasso, we used msgps(): where gamma parameter selection is 1 (*i.e., $\gamma$ in the penalty function*); Initial estimators, as suggested by Zou when $n \gg p$, select the minimum square estimator that meets the requirements of $\sqrt{n}$ convergence $\hat{\beta}_j = \hat{\beta}^{OLS}$ as the initial estimator, corresponding to the parameter $Lambda = 0$; If n<p, $Lambda = 0.001$ as the initial estimate of $\hat{\beta}_j = \hat{\beta}^{ridge}$.

For Elastic-net, we used msgps() where alpha =0.5;

For Lasso, we used msgps() where alpha defaults to 1;

For SCAD, we used local quadratic approximation to obtain the locally optimal solution. We can obtain the Lambda of Lasso, Adaptive Lasso and Elastic-net that used Generalized-Cross-Validationg(*GCV*) to solve the optimal value and obtain the corresponding coefficient value $\hat{\beta}$.

## Numerical simulation result analysis

(1) First examine n:p =2500:50 (*number of samples on the training set: number of predictors*); Number of trial repeats $m =300$, the correlation coefficient is *cor* =0.2/0.4/0.6/0.8/0.98

Numerical simulation obtains Angle, Err boxplot (Fig. 1), the time data (Table 1) and Angle median-correlation coefficient, Err median ∼correlation coefficient relationship curve (Fig. 2).

(2) Comparative analysis. In the context of a large sample size (n ≫p) and without accounting for model stability, this study examines the accuracy of model predictions

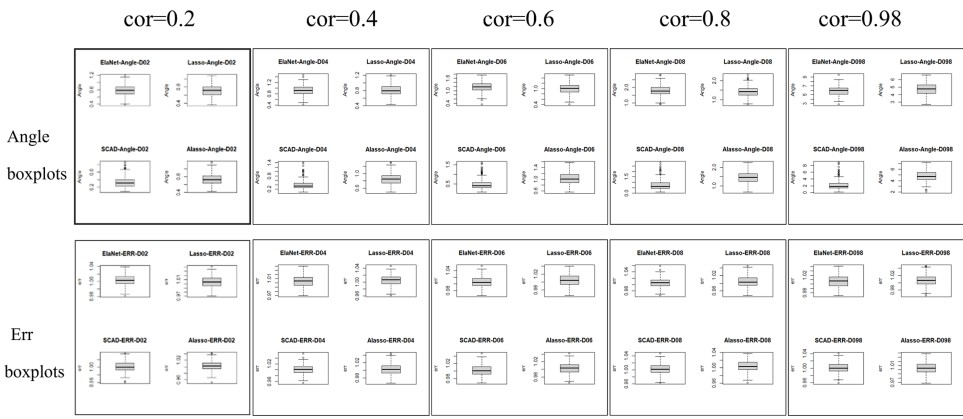

**Figure 1  Angle and err boxplots.**

and the coincidence of model coefficients when the correlation coefficient is less than 0.8. According to the results, SCAD exhibited the highest performance, trailed by Adaptive-Lasso and Lasso. The reason for this is that SCAD is capable of attaining an approximately unbiased estimation of large coefficients through the design of its penalty function. Theoretically, Adaptive-Lasso can improve variable selectivity by weighting corrections to the Lasso penalty terms, However, the results of this numerical simulation experiment show that the two perform basically the same in terms of variable selectivity (For Adaptive Lasso iterative solution of the minimum loss function, *Zou (2006)* proposed that to obtain satisfactory Oracle properties which required an initial estimator $\hat{\beta}^0$ with fully satisfies the $\sqrt{n}$ consistency. It is difficult to achieved whaterver using coordinate descent method or gradient descent method for iterative solution. The models fitted in these four methods all perform well in terms of prediction accuracy (*Err* $\leq 1.005$), where the irreducible error $\varepsilon \sim N(0,1)$ is an upper bound on the prediction accuracy of y, maximum value is 1 (*i.e.*, $\Delta Err \leq 0.005$). In cases where a group of predictors exhibit strong correlation (*i.e.*, cor>0.8), the model's variable selection compatibility Angle experiences a rapid decline. However, the model's Err remains relatively small and is largely unaffected by the increasing impact of the correlation coefficient. Hence, the model constructed amidst the presence of pronounced collinearity among the predictor variables is apt for forecasting the outcome based on the predictor variables. However, it is not conducive for making inferences regarding the association between each predictor variable and the response variable.

The evaluation of model stability and algorithm efficiency can be performed through the utilization of Angle, Err boxplot, and regression fitting timetable Time. The convex penalty function possesses favourable stability properties that can enhance the performance of models constructed through the utilization of Lasso, Elastic-Net, and Adaptive-Lasso techniques.

The Lasso algorithm is characterized by its simplicity and efficiency, which results in a notably high speed of model fitting. Thus, the Lasso method is highly appropriate for modeling in scenarios where the number of observations is greater than the number of predictor variables, and the degree of collinearity among the predictor variables is

| Table 1 | Regression fitting schedule Time (s). | | | | |
|---|---|---|---|---|---|
| Regression method | Time/cor=0.2 | Time/cor=0.4 | Time/cor=0.6 | Time/cor=0.8 | Time/cor=0.98 |
| Elastic-Net | 1.9883 | 2.3013 | 2.2543 | 2.3370 | 2.3300 |
| Lasso | 1.8800 | 2.0140 | 1.9773 | 1.9900 | 2.1473 |
| SCAD | 2.2453 | 2.6367 | 4.1060 | 4.5927 | 12.4660 |
| Adaptive-Lasso | 2.0137 | 1.8079 | 1.9650 | 1.9113 | 1.8607 |

**Notes.**
Computer processor: Intel(R) Core(TM) i7-10710U CPU 1.10 GHz 1.61 GHz.

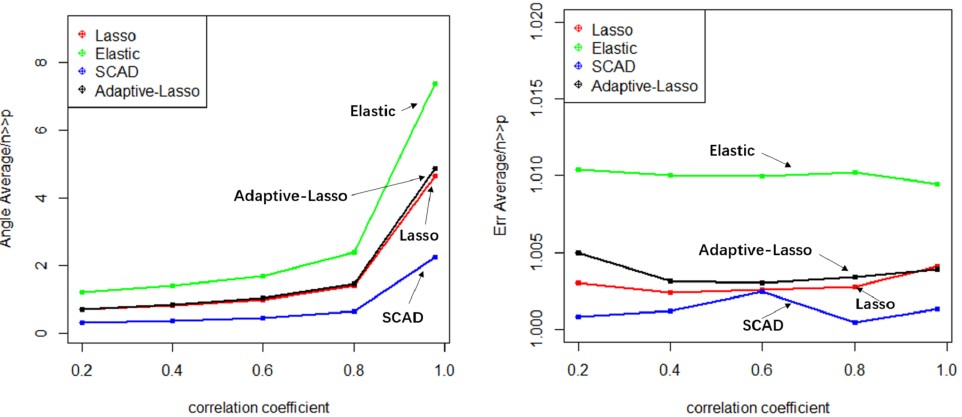

**Figure 2** Relationship graphics.

minimal. Elastic-Net exhibits superior stability under conditions of high correlation coefficients, rendering it appropriate for application scenarios characterized by predictive variables possessing group attributes. The complexity of SCAD's penalty functions, owing to their non-convex nature, results in a model form that is more intricate than that of Lasso, Elastic-Net, and Adaptive Lasso. Additionally, the iterative algorithm employed in SCAD runs notably slower, and the instability of the local quadratic approximation algorithm is evidenced by the Angle and Err in the boxplot with the highest number of outliers. Furthermore, models fitted using this method exhibit poor stability. (3) Next, consider the $n$:$p$ =40:50 (*number of samples on the training set: number of predictors*) high-dimensional dataset; The number of test replicates $m$ =300 and the correlation coefficient was $cor$ =0.2/0. 4/0.6/0.8/0.98. The study conducted a numerical simulation and generated several graphical and tabular outputs, including a boxplot of Angle and Err (Fig. 3), a table of time data (Table 2), and a curve depicting the relationship between the median of Angle-correlation coefficient and Err-correlation coefficient (Fig. 4). (4) Comparative analysis. Upon analyzing the Angle and Err indicators, it can be concluded that SCAD exhibits superior performance. However, when compared to the state of affairs pertaining to a large sample data, the Angle indicator exhibits a significant increase by several to dozens of times. The interpretability of models constructed by the four classical methods is notably weak when dealing with high-dimensional data. The Lasso, SCAD, and

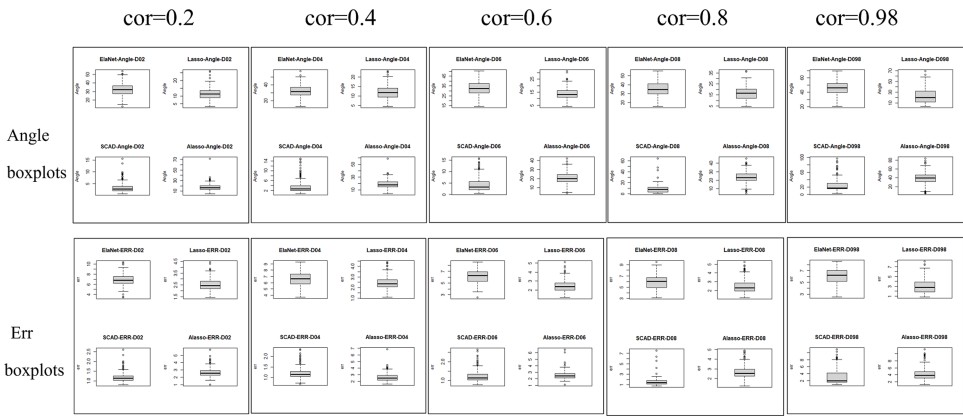

**Figure 3** Angle and err boxplots.

**Table 2** Regression fitting schedule time (s).

| Regression method | Time/cor =0.2 | Time/cor =0.4 | Time/cor =0.6 | Time/cor =0.8 | Time/cor =0.98 |
|---|---|---|---|---|---|
| *Elastic-Net* | 0.334 | 0.328 | 0.358 | 0.396 | 0.502 |
| *Lasso* | 0.286 | 0.285 | 0.293 | 0.293 | 0.345 |
| *SCAD* | 0.135 | 0.141 | 0.154 | 0.156 | 0.201 |
| *Adaptive-Lasso* | 0.308 | 0.312 | 0.362 | 0.338 | 0.410 |

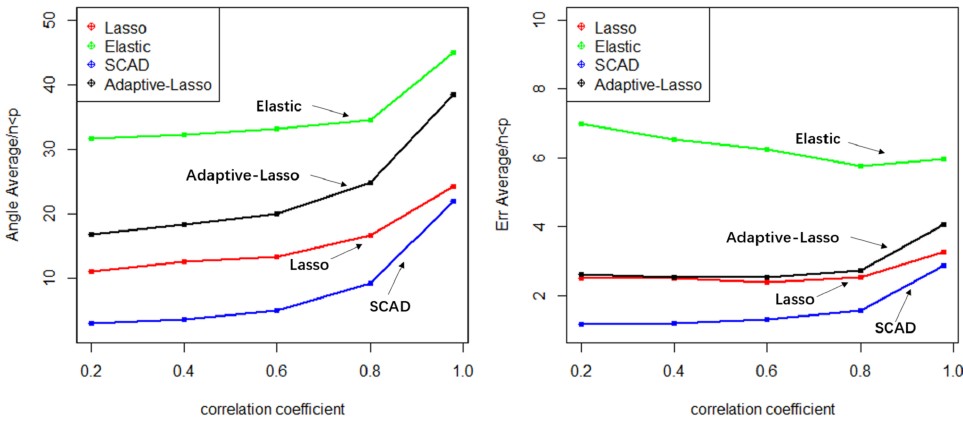

**Figure 4** Relationship graphics.

Adaptive-Lasso variable selection methods have produced models with prediction errors of less than 3. These models may be suitable for application scenarios where high prediction accuracy is not a requirement. In cases where the correlation coefficient between variables

exceeds 0.8, the respective indicators of the model experience a rapid decline, thereby rendering the model's guiding significance null and void.

The unsuitability of the Lasso-like classical variable selection method, which is appropriate for conventional data, for modeling high-dimensional data has been demonstrated through numerical simulation. Over the past decade, there has been a surge in research on high-dimensional models in the field of statistics both domestically and internationally. Numerous modeling theories and application practices have surfaced for various forms of high-dimensional data (*Niu & Li, 2021*). These have proven to be a potent means of managing vast quantities of information during the age of big data.

## Comparative experiment and result discussion

The evaluation of the quality of the fitted model for Lasso, Elastic-Net, Adaptive-Lasso, and SCAD variable selection methods, as per literature, involves the computation of the average of correctly selected predictive variables ("cc"), the average of correctly eliminated unrelated predictive variables ("cd"), and the accuracy of coefficient estimation $\hat{\beta}$ (*Rodgers & Nicewander, 1988*). The evaluation of the efficacy of the fitted model in literature is conducted through a comparison of the test mean square error (PMSE), the count of non-zero coefficients that are erroneously identified as zero ("IC"), and the count of zero coefficients that are correctly identified as zero ("C") (*Yanbiao, 2011*). The literature employs the use of SIS as a means of reducing the dimensionality of high-dimensional data with varying correlation numbers. Subsequently, the fitting process is carried out using Lasso, Adaptive-Lasso, Elastic-Net, and SCAD methods. The resulting regression coefficients $\hat{\beta}$ of each model are then compared to assess the efficacy of the fitting model. All of the findings indicate that the most effective approach for SCAD fitting involves the utilization of non-convex penalty function regression. From a new perspective, this study proposes to calculate the Angle $= \frac{180}{\pi} \arccos(\beta^{\mathrm{T}} \hat{\beta})$ between the unitized estimation coefficient vector $\hat{\beta}$ and the unitized true coefficient vector $\beta$ to evaluate the consistency of variable selection of the fitted model, and analyze with the help of boxplot tool the distribution and stability of Angle. In order to demonstrate the efficacy and practicality of the proposed approach, a comparative study is undertaken between the model evaluation technique presented in this manuscript and conventional analytical methods, under identical conditions.

The linear mixed random effects model is described as $Y = X\beta + Z\delta + \varepsilon$. This study discusses the sparse model. The coefficient vector of the real model is set as $\beta = (1, 1.2, -3.2, 4.1, -5, \underbrace{0, \ldots, 0}_{45})$, in which the number of non-zero coefficients is 5, and the number of zero coefficients is 45. The study conducted numerical simulation experiments in the context of large sample and high dimension. The experiments involved varying correlation coefficients of 0.2, 0.6, and 0.98, and were repeated 300 times. Please refer to Tables 3 and 4 for the obtained outcomes.

Cor: correlation coefficient between predictive variables.

Angle(Median, Outliers): median and number of outliers are given to evaluate respectively the explanatory power and stability of the model

**Table 3 Selection rate/rejection rate (cor=0.6; $m = 300$).**

| Regression method | $n{:}p = 2500{:}50$ | | | | $n{:}p = 40{:}50$ | | | |
|---|---|---|---|---|---|---|---|---|
| | VC | Selection rate | VD | Rejection rate | VC | Selection rate | VD | Rejection rate |
| Elastic-Net | 5 | 100% | 0.093 | ≈ 0 | 5 | 100% | 0.057 | ≈ 0 |
| Lasso | 5 | 100% | 24.5 | 53.8% | 4.98 | 99.6% | 5.99 | 13.3% |
| SCAD | 5 | 100% | 45 | 100% | 4.66 | 93.1% | 43.6 | 96.7% |
| Adaptive-Lasso | 5 | 100% | 26.2 | 58.2% | 4.91 | 98.2% | 4.78 | 10.6% |

**Table 4 Coefficient estimation accuracy (cor=0.6; $m = 300$).**

| Regression method | $n{:}p = 2500{:}50$ Coefficient estimate error | $n{:}p = 40{:}50$ Coefficient estimate error |
|---|---|---|
| Elastic-Net | 0.0135 | 0.474 |
| Lasso | 0.0162 | 0.210 |
| SCAD | 0.0121 | 0.147 |
| Adaptive-Lasso | 0.0134 | 0.288 |

Err (Median , Outliers):(Mean square error of the model)

median and number of outliers are given to evaluate respectively the prediction accuracy and stability of the model

CC: the mean of the number of non-zero coefficients selected

CD: the mean of the number of zero coefficients selected

CEMR: the average error of coefficient estimation $\hat{\beta}$

Table 3 presents the four indicators proposed by this study, namely Angle Median, Angle Outliers, Err Median, and Err Outliers, along with the three commonly used indicators in the research field pertaining to correctly selected predictive variables (represented by "CC"), correctly eliminated irrelevant variables (represented by "CD"), and the average error of coefficient estimation $\hat{\beta}$ (represented by "CEMR"), under the condition of a large sample ($n \gg p$).The numerical simulation results indicate that:
(1) Of the four variable selection methods, namely Lasso, Elastic-Net, Adaptive-Lasso, and SCAD, the latter exhibits superior performance. Specifically, SCAD yields the smallest median angle, which reflects the consistency of variable selection, and achieves a 100% elimination rate for the "cd" value, indicating the removal of irrelevant variables. The model's prediction accuracy, as represented by the median of Err, is at its lowest point, while the estimation error of coefficient $\hat{\beta}$ is minimized. Elastic-Net, akin to Lasso and Adaptive-Lasso, exhibits limited capacity to effectively shrink non-relevant variables.
(2) The performance metrics of Lasso, Elastic-Net, Adaptive-Lasso, and SCAD, which are four variable selection techniques, exhibit a decline with the increase in correlation "cor" among the predictive variables. However, SCAD continues to demonstrate superior performance among the four methods.

(3) The Angle and Err outliers presented in this study can serve as a means to delineate the stability of the model. Empirical evidence confirms the theoretical assertion that the SCAD model's stability is compromised by the non-convexity of its penalty function and the instability of the local quadratic approximation algorithm.

(4) The metric used to evaluate the effectiveness of a model's performance. The present study utilizes median values for Angle and Err, as opposed to mean values. This methodological choice serves to mitigate the influence of outliers on statistical analyses and to enhance the precision of the fitted model's performance evaluation.

Table 4 presents the experimental outcomes of the numerical simulation comparison of two assessment techniques in high-dimensional settings. The analytical results presented in Table 4 are equivalent to those displayed in Table 3 when dealing with sizable sample sizes. Table 4 demonstrates a decline in the indicators of the models acquired through both evaluation methods. When the correlation coefficient (cor) among the predictor variables is significantly high, the performance indicators of each model tend to deteriorate rapidly. This could imply that the model has lost its ability to provide meaningful guidance. The present comparative analysis demonstrates that the outcomes of the two assessment techniques utilized for the four variable selection methods are in agreement. This finding substantiates the efficacy and suitability of the model evaluation method proposed in this study. Additionally, the stability attributes of the model fitted by the four variable selection methods are ascertained by scrutinizing the outliers of Angle and Err indices. In the statistical analysis of Angle and Err, which serve as performance indicators for model inference and prediction ability, the median value is utilized in lieu of the mean value to enhance the precision of the analysis outcomes.

## EMPIRICAL ANALYSIS AND DISCUSSION

Statistical analysis and modeling provide recommendations regarding the most suitable model based on the available data. In order to achieve optimal results, it is imperative to develop a regression model that is both practical and effective, in conjunction with established professional standards. The present study employs the Heart dataset, which comprises 462 observations and nine variables, as made available by R. This dataset pertains to the baseline survey conducted in rural South Africa to investigate coronary risk factors, and is utilized for empirical analysis in this study. The dataset comprises the subsequent variables:

sbp: Systolic blood pressure
tobacco: Cumulative tobacco consumption, in kg
ldl: Low-density lipoprotein cholesterol
adiposity: Adipose tissue concentration
famhist: Family history of heart disease (1 =Present, 0 =Absent)
typea: Score on test designed to measure type-A behavior-
obesity: Obesity
alcohol: Current consumption of alcohol
age: Age of subject

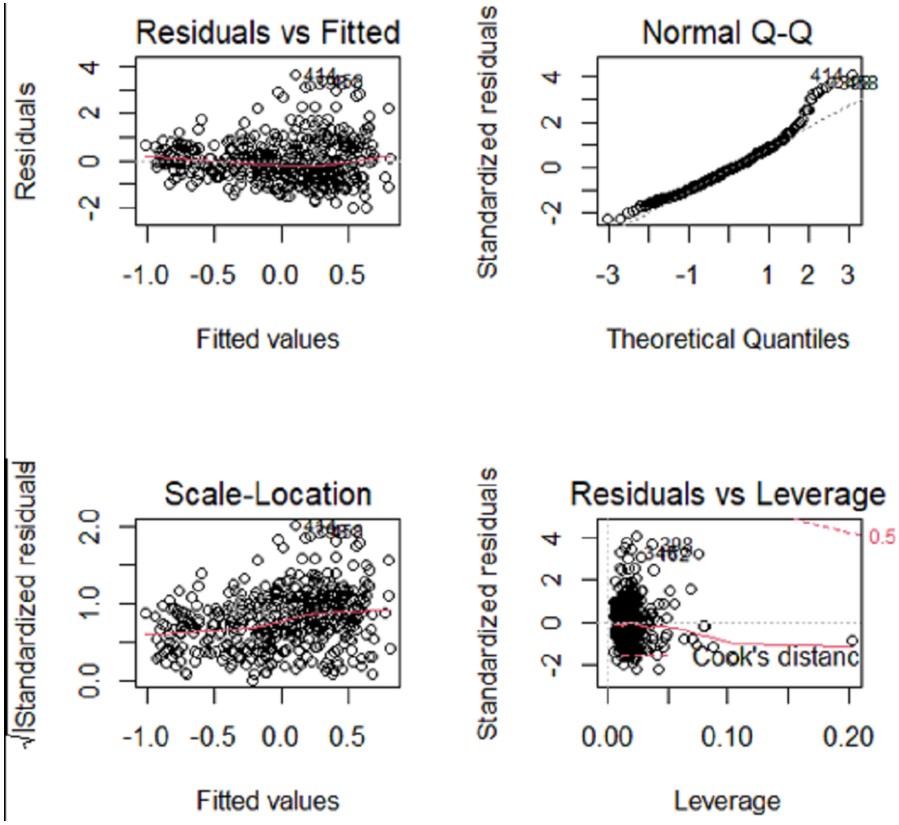

**Figure 5** Sbp regression diagnostic graph.

The dataset is subjected to standardization due to the presence of varying units of measure for each variable. Subsequently, the dataset is partitioned into two distinct subsets, namely the training set and the test set. Various variable selection techniques, including Elastic-Net, Adaptive-Lasso, Lasso, and SCAD, are employed on the training set alongside cross-validation methods to identify the variables that exert a more significant influence on systolic blood pressure and to construct a regression model. The model is validated on the test set, and the mean squared error is computed to assess the efficacy of the models fitted by various techniques.

Initially, the OLS linear regression model was employed to construct a model of the dataset, thereby establishing a linear regression association between systolic blood pressure and other variables. Assess the conformity of the response variable to the statistical assumptions of the linear regression model, including but not limited to normality, independence, homoscedasticity, and linear correlation with the predictor variables. Figure 5 displays the statistical characteristics derived from conducting OLS regression on systolic blood pressure (sbp) in the Heart dataset.

The QQ plot, Residuals *vs* Fitted plot, and Scale-Location plot suggest that the response variable sbp adheres to the assumptions of normality, linearity, and homoscedasticity. However, independence cannot be discerned from the visualizations. There is no inherent

**Table 5  Heart: OLS regression model coefficients.**

| (*Intercept*) | *tobacco* | *ldl* | *adiposity* | *famhist* | *typea* | *obesity* | *alcohol* | *age* |
|---|---|---|---|---|---|---|---|---|
| −0.0313 | −0.0251 | 0.0317 | 0.1329 | −0.0953 | −0.052 | 0.0908 | 0.0846 | 0.3016 |

**Figure 6  Elastic-Net parsing path and cross-validation MSE graph.**

justification to believe that the systolic blood pressure of an individual has an impact on another person's systolic blood pressure. Therefore, it is rational to presume that the systolic blood pressure also conforms to the principle of independence.

Initially, conduct OLS linear regression analysis on the Heart dataset's training subset to establish a linear functional association between systolic blood pressure and diverse influencing factors. Table 5 displays the estimated coefficients values.

The accuracy of the model was evaluated on the test set by computing the mean square error of the index, resulting in an error value of 0.92. The OLS model posits that there exists a relationship between the response variable and all predictors, albeit the response variable is typically associated with a subset of the predictor variables. Consequently, to attain variable selection and enhance prediction accuracy, the present study employs the Lasso regression approach to construct the model and perform comparative analysis.

## Elastic-Net

The parsing path of the Elastic-Net penalty function and the cross-validation mean squared error graph are depicted in Fig. 6. The R programming language's in-built ten-fold cross-validation function was utilized to determine the optimal adjustment parameter λ value of 0.107, which corresponds to the smallest training mean square error. Subsequently, an Elastic Net regression model was fitted using this value, and the resulting regression coefficients are presented in Table 6.

The model was validated on the test set and the mean square error of the index, which serves as a measure of the model's accuracy, was calculated to be Err =0.926. The study aimed to perform variable selection and identify the factors that significantly affect systolic

**Table 6  Heart: elastic-net-coefficients.**

| (*Intercept*) | *tobacco* | *ldl* | *adiposity* | *famhist* | *typea* | *obesity* | *alcohol* | *age* |
|---|---|---|---|---|---|---|---|---|
| 0.0333 | 0.0560 | | 0.1815 | | | 0.0127 | 0.0335 | 0.1372 |

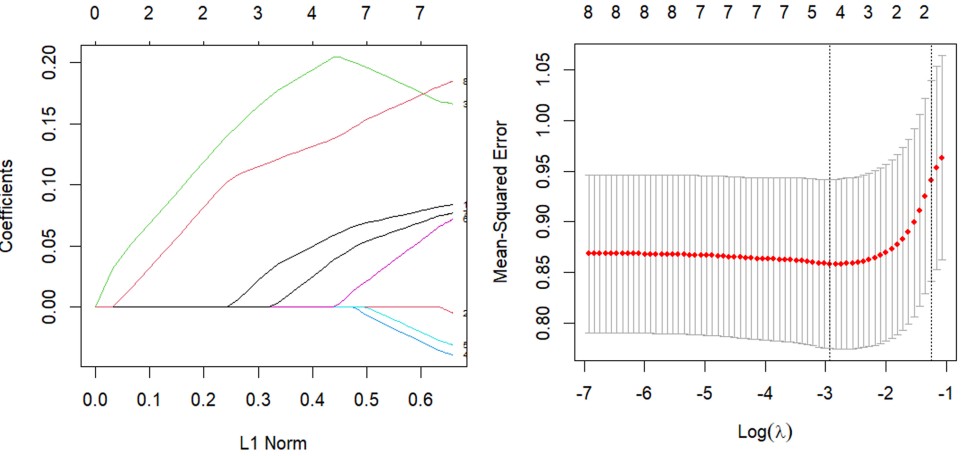

**Figure 7  Lasso parsing path and cross-validation MSE graph.**

**Table 7  Heart: Lasso-coefficients.**

| (*Intercept*) | *tobacco* | *ldl* | *adiposity* | *famhist* | *typea* | *obesity* | *alcohol* | *age* |
|---|---|---|---|---|---|---|---|---|
| 0.0332 | 0.0561 | | 0.2015 | | | | 0.0347 | 0.1359 |

blood pressure, including tobacco consumption, adipose tissue concentration, obesity, alcohol consumption, and age. The quantitative relationship between these factors was established, with age and adipose tissue concentration found to have the most significant impact.

## Lasso

The parsing path of the Lasso penalty function and the cross-validation mean squared error graph are depicted in Fig. 7. The R language's in-built ten-fold cross-validation function was utilized to determine the optimal adjustment parameter $\lambda = 0.0536$, which yielded the smallest training mean square error. Subsequently, a Lasso regression model was fitted using this parameter, and the resulting regression coefficients are presented in Table 7. The model should be validated on the test set and the mean square error of the index, which serves as an indicator of the model's accuracy, should be calculated. The value of Err, which is equal to 0.929, should be obtained. Additionally, variable selection should be performed. In contrast to the Elastic-Net approach, the removal of the obesity term and the augmentation of the impact factor of adipose tissue concentration on systolic blood pressure not only streamlines the model, but also possesses a logical basis, as evidenced by the correlation coefficient of 0.72 obtained from the corr.test() function in the psych package when analyzing the relationship between the obesity term and adipose tissue

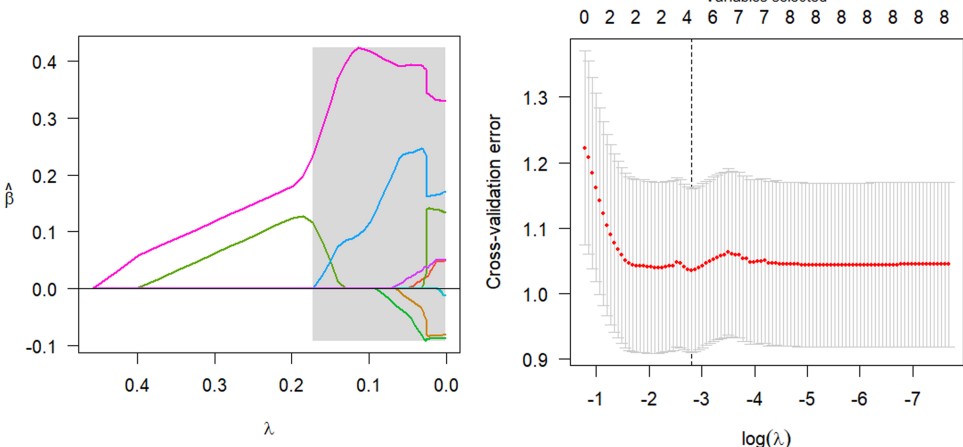

**Figure 8** SCAD parsing path and cross-validation MSE graph.

**Table 8** Heart: SCAD- coefficients.

| (*Intercept*) | *tobacco* | *ldl* | *adiposity* | *famhist* | *typea* | *obesity* | *alcohol* | *age* |
|---|---|---|---|---|---|---|---|---|
| 0.0225 | | −.0053 | | −0.0351 | | 0.2288 | 0.0081 | 0.3915 |

concentration. The process of parsing exhibits a comparable pattern, and demonstrates a higher rate of compression as the coefficient experiences an increase in lambda.

## SCAD

The parsing path of the SCAD penalty function is illustrated in Fig. 8. The region that is shaded in the diagram signifies the attainment of a local optimum by SCAD within that particular area, which consequently leads to the emergence of an unstable solution defect. Utilize the R programming language's inherent cross-validation function, specifically the ten-fold cross-validation method, to derive the adjustment parameter $\lambda = 0.149$ at the point of minimal training mean square error. Subsequently, generate the SCAD fitting regression model. Table 8 displays the regression coefficients that correspond to the given data.

The model should be validated on the test set, and the mean square error of the index that reflects the model's accuracy, Err =0.946, should be calculated. Additionally, variable selection should be performed. The study identified and isolated five key factors that exhibited a significant influence on systolic blood pressure. Additionally, the quantitative relationship between these factors was elucidated. Nevertheless, the solution provided by SCAD exhibits instability, and the assigned coefficient of ldl is negative, a result that lacks plausibility.

## Adaptive Lasso

The parsing path of the Adaptive Lasso penalty function is illustrated in Fig. 9. The msgps package provides adjustment parameters for various variable selection criteria, including

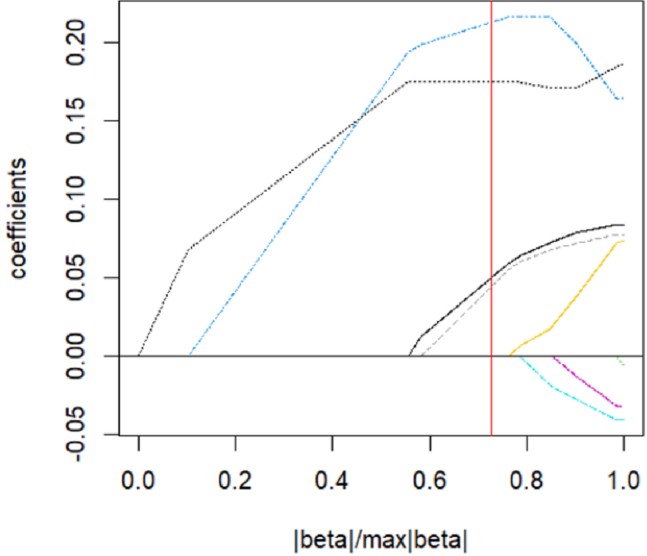

**Figure 9   Adaptive Lasso parsing path.**

**Table 9   Heart: Adaptive Lasso-coefficients.**

| (*Intercept*) | *tobacco* | *ldl* | *adiposity* | *famhist* | *typea* | *obesity* | *alcohol* | *age* |
|---|---|---|---|---|---|---|---|---|
| 0.0305 | 0.0506 | | 0.2132 | | | | 0.0450 | 0.1754 |

**Table 10   Comparison of mean square errors by models.**

| Method | Err |
|---|---|
| *Elastic-Net* | 0.926 |
| *Lasso* | 0.929 |
| *SCAD* | 0.946 |
| *Adaptive-Lasso* | 0.920 |

AIC, BIC, GCV, and Cp. The present study has opted for the GCV (ms.tuning = 2.86) approach to derive an Adaptive Lasso regression model with a good fit. Table 9 displays the regression coefficients that correspond to the given data.

The model is validated on the test set, and the mean square error of the index is calculated to reflect the accuracy of the model, resulting in an error of 0.92. This indicates that variable selection has been successfully achieved. Table 10 demonstrates the presented information. The concentration of adipose tissue, age, tobacco and alcohol consumption are identified as significant factors that exert a greater influence on systolic blood pressure. Further, the quantitative relationship between these factors is refined for better understanding.

Result analysis:

(1) The Lasso technique is effective in achieving variable selection and identifying the factors that exert a significant influence on systolic blood pressure. In contrast to the

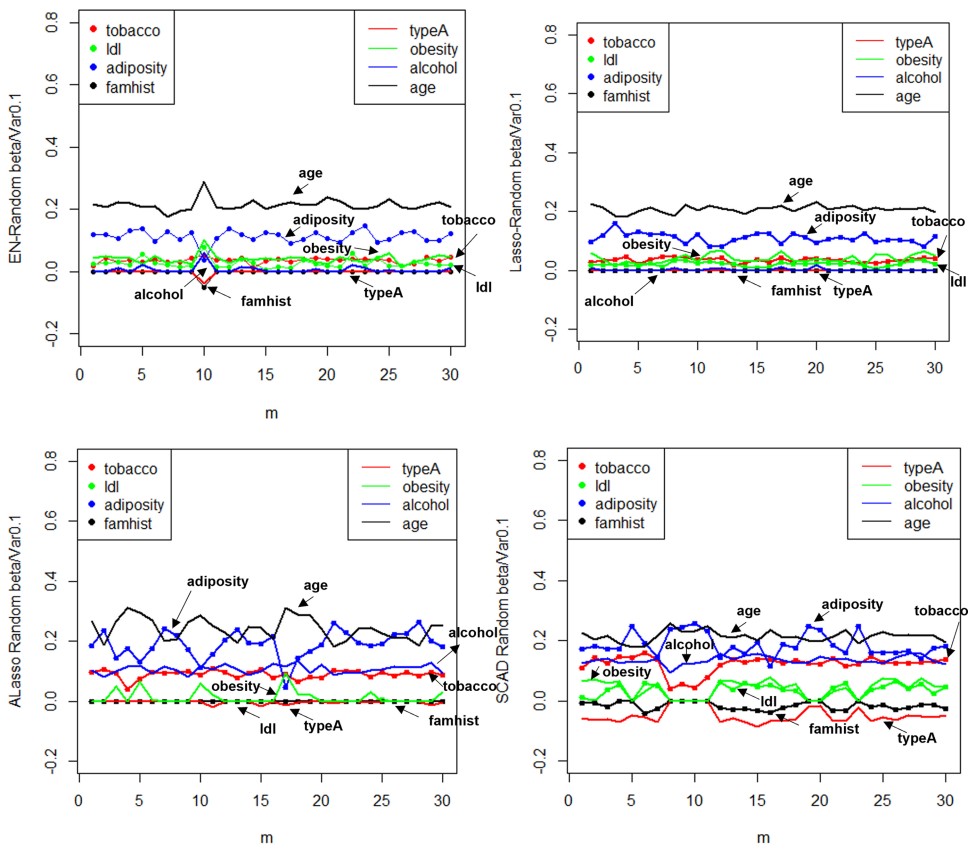

**Figure 10** $\hat{\beta}$ random effect diagram.

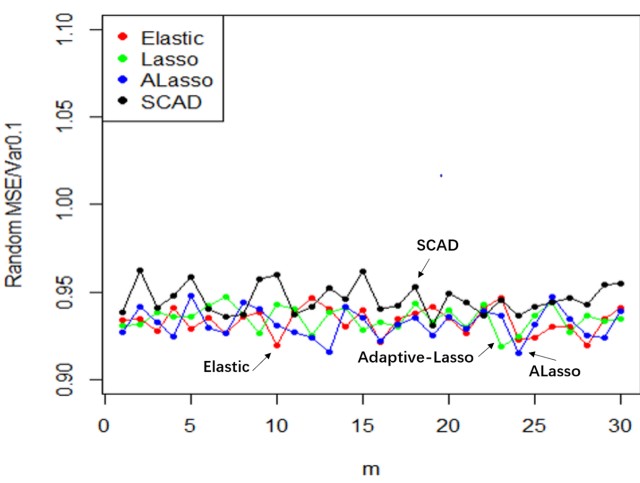

**Figure 11** MSE random effect diagram.

Elastic-Net approach, the exclusion of the obesity variable and the augmentation of the determinants of adipose tissue concentration on systolic blood pressure are observed. The model can be deemed as straightforward and rational, as evidenced by the correlation coefficient of 0.72 between the obesity term and the adipose tissue concentration. The process of parsing exhibits a likeness as well, and experiences expedited compression as the coefficient ascends in tandem with lambda.

(2) The instability of the solution provided by SCAD is evidenced by the negative coefficient assigned to ldl, which is deemed implausible.

(3) Similar to Lasso, the Adaptive Lasso method effectively eliminated adipose tissue concentration, age, tobacco, and alcohol as variables that exert a significant influence on systolic blood pressure. Moreover, the Adaptive Lasso approach yielded the lowest mean square error.

The utilization of Elastic-Net, Lasso, Adaptive Lasso, and SCAD has been employed in fitting the random effect of the coefficient estimate $\hat{\beta}$ depicted in Fig. 10, as well as the random effect of the mean square error illustrated in Fig. 11, based on the given design. To examine and evaluate the stochastic impact of $\hat{\beta}$, it is possible to observe. The model developed through the utilization of Adaptive Lasso and SCAD exhibits a marginally superior capacity for variable selection in comparison to Elastic-Net and Lasso. The model coefficients are subject to significant fluctuations under the influence of random effects, whereas Elastic-Net and Lasso exhibit relatively stable behavior. Both the mean square error and stability of each model are similarly impacted by random factors. Thus, during the practical implementation phase, the most suitable technique for variable selection can be chosen to align with the model requirements.

## CONCLUSION

The COVID-19 pandemic remains a significant global concern, as evidenced by data obtained from public platforms and the Lasso class variable selection method described in our study. We utilized a linear random effects model to examine the interrelationships among various factors, including COVID-19 infection rates, mortality rates, population density, urban population, proportion of individuals aged 65 and older, vaccination rates, per capita GDP from the previous year, number of hospital beds per thousand people, human development index, proportion of patients with underlying diseases, and overall government response index, among others. This information can aid the government in comprehending the extent of COVID-19 transmission, allowing for the prompt development and modification of epidemic prevention and control policies.

### Funding

This research was supported by the Wenzhou Municipal Basic Scientific Research Project (R20220001), the Key Project of Philosophy and Social Science Research in Zhejiang Province (23NDJC045Z), the Wenzhou Philosophy and Social Science Planning Project

(22WSK305), the Research project of Zhejiang Education Department (Y202249099), and the General Project of the National Social Science Foundation (22BJY126). The funders had no role in study design, data collection and analysis, decision to publish, or preparation of the manuscript.

### Grant Disclosures

The following grant information was disclosed by the authors:
Wenzhou Municipal Basic Scientific Research Project: R20220001.
Key Project of Philosophy and Social Science Research in Zhejiang Province: 23NDJC045Z.
Wenzhou Philosophy and Social Science Planning Project: 22WSK305.
Research project of Zhejiang Education Department: Y202249099.
General Project of the National Social Science Foundation: 22BJY126.

### Competing Interests

The authors declare there are no competing interests.

### Author Contributions

- Dake Hou conceived and designed the experiments, performed the computation work, prepared figures and/or tables, authored or reviewed drafts of the article, and approved the final draft.
- Wenli Zhou performed the experiments, prepared figures and/or tables, and approved the final draft.
- Qiuxia Zhang analyzed the data, authored or reviewed drafts of the article, and approved the final draft.
- Kun Zhang analyzed the data, authored or reviewed drafts of the article, and approved the final draft.
- Jiaqi Fang conceived and designed the experiments, authored or reviewed drafts of the article, and approved the final draft.

### Data Availability

 The code is available in the Supplemental Files.

### Supplemental Information

Supplemental information for this article can be found online at http://dx.doi.org/10.7717/peerj-cs.1522#supplemental-information.

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
