# Peer review of "A comparative study of different variable selection methods based on numerical simulation and empirical analysis"

_PeerJ Computer Science, doi:10.7717/peerj-cs.1522_

## Round 0.1 · original submission · Major Revisions

The experimental design could be improved by increasing the breadth and depth of the comparative experiments. A more comprehensive set of comparative experiments will provide a more robust evaluation of the proposed method and enhance the persuasiveness of the research. By demonstrating the effectiveness of the proposed method under various conditions, the author can establish its validity and applicability. Please address this issue during the revision process and consider augmenting the experimental design with additional comparative experiments to enrich the data and strengthen the manuscript.

Reviewer 1 ·

Basic reporting

1. Authors should emphasize contribution and novelty in the abstract. Moreover, the introduction needs to clarify the (1) motivation, (2) challenges, (3) contribution, (4) objectives, and (5) significance/implication.
2. It helps to appreciate the paper by having a related review section. The authors should consider more recent research done in the field of their study (especially in the year 2021 and 2022 onwards). If possible, the authors can give a table pinpointing the advantage or limitations of each work.
3. The authors need to interpret the meanings of the variables. Some parameters and their values are unknown. It would be better to show all these parameters and explain the reason for those numbers in the table.

Experimental design

4. The manuscript requires further discussion and more details analysis. The authors should provide enough proof to convince the reader of superiority of the proposed schemes over the existing works.

Validity of the findings

5. The conclusion and future work part can be extended to have a better understanding of the approach and issues related to that which can be taken into consideration for future work.

Additional comments

1. Authors should emphasize contribution and novelty in the abstract. Moreover, the introduction needs to clarify the (1) motivation, (2) challenges, (3) contribution, (4) objectives, and (5) significance/implication.
2. It helps to appreciate the paper by having a related review section. The authors should consider more recent research done in the field of their study (especially in the year 2021 and 2022 onwards). If possible, the authors can give a table pinpointing the advantage or limitations of each work.
3. The authors need to interpret the meanings of the variables. Some parameters and their values are unknown. It would be better to show all these parameters and explain the reason for those numbers in the table.
4. The manuscript requires further discussion and more details analysis. The authors should provide enough proof to convince the reader of superiority of the proposed schemes over the existing works.
5. The conclusion and future work part can be extended to have a better understanding of the approach and issues related to that which can be taken into consideration for future work.

Reviewer 2 ·

Basic reporting

To improve the novelty and impact of the research, it is suggested that the author increases the number of references from the past three years. This will demonstrate the currency of the research and its contribution to the current state of knowledge in the field. The inclusion of more recent literature will also enhance the relevance of the research. Please address this issue during the revision process and consider adding more recent references to strengthen the manuscript.

Experimental design

The experimental design could be improved by increasing the breadth and depth of the comparative experiments. A more comprehensive set of comparative experiments will provide a more robust evaluation of the proposed method and enhance the persuasiveness of the research. By demonstrating the effectiveness of the proposed method under various conditions, the author can establish its validity and applicability. Please address this issue during the revision process and consider augmenting the experimental design with additional comparative experiments to enrich the data and strengthen the manuscript.

Validity of the findings

The Discussion section is a crucial component of the manuscript where the author can offer a critical evaluation of the research and its implications.It is suggested that the author expand the Discussion section by providing a more comprehensive analysis and interpretation of the results. By providing a more detailed and nuanced discussion, the author can demonstrate a firm grasp of the research and its significance. Therefore, please address this issue during the revision process and consider augmenting the content in the Discussion section to enrich the manuscript.

Additional comments

1. The manuscript highlights the importance of variable selection in the model building process, and presents an innovative approach to improving consistency in variable selection. Additionally, the study compares four commonly used classic variable selection methods (Lasso, Elastic-Net, Adaptive Lasso, and SCAD) through numerical simulation and empirical analysis. While the study offers valuable insights into this area, the innovation of the research may be limited. It is recommended that the author provide a more detailed discussion of the current limitations of existing methods or models in this area, and explain how the proposed approach addresses these limitations in a novel way. Additionally, the author may want to consider providing more concrete examples or case studies that demonstrate the practical applications of the proposed approach, and how it offers advantages over existing methods in real-world settings.

2.Can Tables 5 through 9 in this article be combined into one table?Additionally, it is unclear whether there are further data analyses that could be conducted to provide a more robust understanding of the findings.

3.The manuscript includes several images, such as Figures 6, 7, and 10, where the screenshot size is inconsistent. This inconsistency affects the visual appeal of the manuscript and may make it harder for readers to interpret the images. It is recommended that the author adjust the size of these images to make them consistent and improve the overall aesthetic quality of the manuscript. Additionally, the author may want to consider using image editing software to crop or resize the images for greater consistency and clarity.

---

## Round 0.2 · Minor Revisions

Please address the remaining comments.

Reviewer 2 ·

Basic reporting

1.I suggest that you make changes to the format of the in-text citations. It is recommended to place them in the upper right corner of the text to comply with the academic paper formatting requirements. Please carefully check the format of each cited reference while making the changes, ensuring that they are consistent with the citation style you are using, in order to avoid any formatting errors.

2. I suggest that you unify the formatting of the formulas used in the text and ensure that the numbering on the right side is consistent. Please carefully check each formula and ensure that they are formatted in the same way throughout the paper, and that the numbering is consistent on the right side of each formula.

Experimental design

no comment

Validity of the findings

no comment

Additional comments

no comment

---

## Round 0.3 · accepted · Accept

The authors have addressed all comments.